# Neutral Forces and Balancing Selection Interplay to Shape the Major Histocompatibility Complex Spatial Patterns in the Striped Hamster in Inner Mongolia: Suggestive of Broad-Scale Local Adaptation

**DOI:** 10.3390/genes14071500

**Published:** 2023-07-22

**Authors:** Pengbo Liu, Guichang Li, Ning Zhao, Xiuping Song, Jun Wang, Xinfei Shi, Bin Wang, Lu Zhang, Li Dong, Qingduo Li, Qiyong Liu, Liang Lu

**Affiliations:** 1National Key Laboratory of Intelligent Tracking and Forecasting for Infectious Diseases, National Institute for Communicable Disease Control and Prevention, Chinese Center for Disease Control and Prevention, Beijing 102206, China; lpbccc@sina.com (P.L.); zhaoning@icdc.cn (N.Z.); liuqiyong@icdc.cn (Q.L.); 2School of Public Health, Cheeloo College of Medicine, Shandong University, Jinan 250012, China; 3Public Health School, Jiamusi University, Jiamusi 154007, China

**Keywords:** local adaptation, major histocompatibility complex, balancing selection, parasite, striped hamster, population differentiation

## Abstract

Background: The major histocompatibility complex (MHC) plays a key role in the adaptive immune response to pathogens due to its extraordinary polymorphism. However, the spatial patterns of MHC variation in the striped hamster remain unclear, particularly regarding the relative contribution of the balancing selection in shaping MHC spatial variation and diversity compared to neutral forces. Methods: In this study, we investigated the immunogenic variation of the striped hamster in four wild populations in Inner Mongolia which experience a heterogeneous parasitic burden. Our goal was to identify local adaptation by comparing the genetic structure at the MHC with that at seven microsatellite loci, taking into account neutral processes. Results: We observed significant variation in parasite pressure among sites, with parasite burden showing a correlation with temperature and precipitation. Molecular analysis revealed a similar co-structure between MHC and microsatellite loci. We observed lower genetic differentiation at MHC loci compared to microsatellite loci, and no correlation was found between the two. Conclusions: Overall, these results suggest a complex interplay between neutral evolutionary forces and balancing selection in shaping the spatial patterns of MHC variation. Local adaptation was not detected on a small scale but may be applicable on a larger scale.

## 1. Introduction

The term “local adaptation” refers to the phenomenon when a population evolves to be better adapted to its specific local environment compared to migrants from nearby populations [1]. Adaptation is crucial for the survival of populations in dynamic environments characterized by ongoing changes such as climate change, anthropogenic influences, and emerging infectious diseases. The capacity of populations to adapt to spatially diverse and temporally fluctuating landscapes is influenced by the interplay of natural selection, gene flow, and genetic drift [2,3,4]. Therefore, examining genetic similarities and differences within and between populations not only unveils the evolutionary history of a species but also offers valuable insights into its future evolutionary potential. Historically, numerous studies have concentrated on habitat adaptation, trophic specialization, and the impact of resource competition [5]. However, these factors not only directly impact organism performance but also have indirect effects on other ecosystem components that can influence overall performance [6].

In host–parasite systems, the genetic variation among hosts and parasites creates dynamic environments that require adaptation. The degree of local adaptation is influenced by gene flow, as well as the variation in genes associated with fitness. Studies on host–parasite adaptation often prioritize examining the ability of pathogens to adapt to their hosts, as the shorter generation times of pathogens make them more likely to exhibit patterns of local adaptation. However, parasites can engender significant ecological and evolutionary interactions with both hosts and host populations [7]. Parasites thrive at the expense of other organisms by extracting energy from their hosts, potentially leading to reduced host condition and reproductive success [8,9]. Specifically, infections can initiate and facilitate local immunogenic adaptation, which relies on dissimilarities within the parasite community [10]. The geographical distribution of parasites can be highly heterogeneous due to various ecological factors. This distribution shapes the genotype distribution of their hosts through natural selection and contributes to host genetic diversity through balancing and disruptive selection [11,12].

The major histocompatibility complex (MHC) plays a crucial role in adaptive immunity, primarily due to its ability to recognize antigens and initiate the immune response [13]. MHC class I presents antigens of intracellular parasites to cytotoxic T-cells, while MHC class II presents antigens of extracellular parasites to helper T-cells [14]. As antigen binding is highly specific, MHC genes exhibit considerable diversity, even in small or bottlenecked populations [15,16,17]. Large numbers of alleles have been identified in various vertebrate species, including rodents [18,19,20,21]. Three major hypotheses, namely Heterozygote Advantage (HA), Negative Frequency Dependent Selection (NFDS), and Fluctuating Selection (FS), have been proposed to account for the high levels of polymorphism [22]. Importantly, due to the adaptive significance and enduring persistence of MHC polymorphism, the MHC has emerged as a paradigm for studying pathogen-mediated balancing selection [23]. Spatial variation in the selection of MHC genes can result in a mosaic of immunogenic divergence and local adaptation across populations [24,25]. Several studies have explored how balancing selection acts on MHC alleles in response to variations in local parasite-related selective pressures [15,26,27,28].

The striped hamster (*Cricitulus barabensis*) frequently causes significant agricultural damage and has the potential to act as a vector for numerous severe zoonotic diseases [29]. This species is solitary and leaves its birthplace to breed after reaching sexual maturity. The striped hamster is widely distributed in the open landscapes of southern Siberia, Mongolia, and northern China. Previous studies have classified the species into several subspecies based on phenotype and genotype [30], revealing the significant potential for differentiation and local adaptation under varying environmental conditions. Due to its susceptibility to various parasites and the establishment of several laboratory models for studying parasite infections [31,32], the striped hamster serves as an ideal model for investigating local adaptation mediated by parasites. 

The objective of our study was to analyze the spatial patterns of the immune locus MHC and neutral microsatellite markers across four sympatric populations of striped hamsters in order to identify patterns of local adaptation. First, we anticipated discovering the neutral history that has influenced these populations using microsatellite data. Second, we employed high-throughput sequencing to assess the sequence diversity of MHC DRB exon 2 and identified divergence among populations. Finally, in order to assess the impact of balancing selection on MHC genetic variation, we conducted a comparison of genetic diversity and structuring at microsatellite loci and MHC. Our hypothesis suggests that varying parasitic burdens may induce spatial diversity in the MHC among different populations of striped hamsters in order to facilitate adaptation within their local environments. Our research will contribute to a deeper understanding of the patterns of local adaptation and provide essential data for the development of novel public health surveillance strategies in Inner Mongolia.

## 2. Materials and Methods

### 2.1. Ethics Statement

This study received ethical approval from the Ethical Committee of the National Institute for Communicable Disease Control and Prevention, Chinese Center for Disease Control and Prevention. All rodents were captured in areas designated for rodent control, which were neither privately owned nor protected. The described field studies did not require any specific permits.

### 2.2. Study Area and Sampling Methods

The study was conducted in the grasslands of the Inner Mongolia Autonomous Region, China. The region extends over a vast area of 2400 km from east to west, with a precipitation gradient difference of 300 mm per year. It also has a maximum north–south distance of 1700 km, and the annual temperature difference exceeds 10 °C. To ensure the selection of optimal sampling locations for effective inter-site comparisons, we carefully chose nine sampling points based on prior sampling experience. We considered both the east–west direction (precipitation gradient) and the north–south direction (temperature gradient) (Figure 1, Appendix A). Temperature and precipitation data were obtained from Worldclim at a resolution of 5 km² [33].

We conducted a total of six sampling events over a two-year period, which took place three times per year in May (spring), July (summer), and September (autumn). We deployed a total of 600 traps, covering an area of approximately 0.05 square kilometers during each sampling event. To ensure sufficient sample sizes for each population, we left the traps in place for one to two days at each location. To minimize potential parasite influence, we placed each captured striped hamster in a clean cloth bag every morning after checking the traps. The bags were securely fastened to prevent parasites from escaping. The impact of dead hamsters on parasite count was relatively consistent among individuals, and therefore did not significantly affect the final results.

### 2.3. Parasite Screening

We meticulously extracted ectoparasites from the hamsters’ fur using a comb. We meticulously and comprehensively enumerated fleas and gamasid mites for each specimen. We microscopically identified parasites to the species level based on morphological characteristics. Our focus was on the average parasite burden per individual at the population level. To account for the low population sizes in certain groups, we chose a subset consisting of N2, N3, W2, and W4 for comparing parasite pressure across populations using the Kruskal–Wallis test in R 4.1.1. 

### 2.4. Microsatellite Genotyping

We collected multiple tissues, including lung, liver, spleen, and kidney, from each individual. Before DNA extraction, all tissue samples were rapidly frozen in liquid nitrogen and stored at −80 °C. Total genomic DNA was extracted from liver samples using the Qiagen DNeasy Blood and Tissue Kit, following the manufacturer’s instructions. We genotyped the total genomic DNA extracted from each individual using a set of 7 highly polymorphic microsatellite markers (Appendix A). The forward primers of the markers were labeled with four different fluorophores (FAM, HEX, ROX, and TAMRA) at the 5′ end. PCR was performed with a final volume of 25 μL, consisting of 12.5 μL of 2× PCR mix, 1 μM concentration of each primer, 1 μL of DNA, and 9.5 μL of ddH2O. The PCR conditions were as follows: 10 min at 94 °C, followed by 30 touchdown cycles of 30 s at 94 °C, 30 s at 64 °C (−0.5 °C per cycle), and 30 s at 72 °C, 10 cycles of 30 s at 94 °C, 30 s at 49 °C, and 30 s at 72 °C, and a 10 min final extension step at 72 °C. The amplified products were genotyped on an ABI 3730xl Genetic Analyzer, and the output was analyzed using GeneMarker software [34].

### 2.5. Microsatellite Analysis

We assessed the genetic diversity of each locus and population using GenAlEx [35]. The following parameters were utilized: number of alleles (*Na*), number of effective alleles (*Ne*), Shannon’s information index (*I*), observed heterozygosity (*Ho*), expected heterozygosity (*He*), and Fixation index (*F*). [36]. We assessed linkage disequilibrium (LD) and deviations from the Hardy–Weinberg equilibrium (HWE) using the Fisher’s exact test in Arlequin [37]. 

We conducted a series of analyses to identify the genetic structure among populations of striped hamsters. Initially, we employed a Bayesian-model-based algorithm implemented in STRUCTURE [38] to determine the probable number of clusters. We performed ten independent runs for each K value (K = 1 to 4), with a ‘burn-in’ period of 50,000 iterations and 1,000,000 replications. To determine the most likely number of clusters, we employed Evanno’s method implemented in Structure Harvester [39]. The results of ten replicate runs for each K value were aggregated using the Greedy algorithm of Clumpp [40]. Graphical representations of the summary outputs were generated using District [41]. Additionally, principal component analysis (PCA) was conducted using GenAlEx to assess population structure at the MHC-DRB locus and maximize intergroup variation.

### 2.6. MHC Genotyping

We targeted the MHC class II DRB gene exon 2, as it harbors the majority of functionally important antigen binding sites that have been extensively studied [42]. Amplification of a 171 bp fragment of DRB exon 2 was achieved using modified primers [43]. The PCR employed fusion primers that incorporated Illumina adapters and a 6bp barcode, ensuring unique molecular tagging for each individual through distinct barcode combinations across the forward and reverse fusion primers. We used unique barcode combinations across forward and reverse fusion primers so that each individual had a unique molecular tag. To ensure accurate genotyping, three independent amplifications were performed for each individual, allowing for comprehensive evaluation. The PCR reaction was carried out for 35 cycles with temperature cycles of 94 °C for 30 s, 55 °C for 30 s, and 72 °C for 30 s. Following visualization of the amplicons on 1% agarose gels, equimolar pooling was performed to create sequencing libraries. We used qPCR method to quantify our libraries prior to sequencing. Sequencing was performed using two Illumina MiSeq 2 × 150 bp runs (Santiago, CA, USA) following purification and dilution.

Subsequently, raw sequence data was processed into individual genotypes using the Stepwise Threshold Clustering program implemented in the amplisas web software [44]. The software generates a table presenting individual genotypes (columns) and unique MHC sequences (rows) along with their respective read depths. A maximum read depth of 5000 was set for each individual, ensuring the capture of all possible MHC variants. Clustering parameters were set according to Illumina data recommendations, including 1% substitution errors, 0.001% indel errors, and a minimum dominant frequency of 25%. To ensure the analysis of genuine MHC alleles, we excluded sequences containing stop codons and retained only those alleles recovered in at least two distinct individuals.

### 2.7. MHC Analysis

#### 2.7.1. Test for Positive Selection

To assess the historical selection pressure on DRB, we calculated the relative ratio of non-synonymous (*dN*) to synonymous (*dS*) base pair substitutions (ω) using the Nei and Gojobory method [45], with the Jukes–Cantor correction applied. We performed an overall average Z-test of selection implemented in MEGA to examine the differences in ω rates. Two approaches were employed to detect the signature of positive selection for each codon. Firstly, we tested seven models in EasyCodeML software [46], which accounted for different selection intensities among sites: M0 (one-ratio), M1a (nearly neutral), M2a (positive selection), M3 (discrete), M7 (beta distribution), M8 (beta distribution and positive selection), and M8a (beta distribution and nearly neutral selection). The Bayes empirical Bayes (BEB) procedure was used to estimate the Bayes Posterior Probability (BPP), with BPP values exceeding 95% considered significant. A Bayesian inference analysis using MrBayes [47] was conducted to generate the phylogenetic tree in the required input format (nekwick). The best-fit model for the Bayesian inference analysis was determined using jModelTest based on the Akaike Information Criterion (AICs) [48]. The likelihood ratio tests (LRT) in CODEML were employed to compare the codon-based models. Additionally, the Hyphy package in the DataMonkey web server was used to detect the signature of positive selection for each codon, utilizing four different codon-based maximum likelihood tests: fixed effects likelihood (FEL), branch-site unrestricted statistical test for episodic diversification (BUSTED), fast unconstrained Bayesian approximation (FUBAR), and the mixed effects model of evolution (MEME) [49]. 

#### 2.7.2. MHC Diversity, Polygenetic Relationships, and Population Structure

Vertebrates commonly exhibit variation in the number of MHC loci within individuals, posing challenges in assigning MHC alleles to specific loci. To assess MHC allelic diversity at the intra-population level, we quantified the number of alleles per population (*A*), the number of segregating sites (S), and nucleotide diversity (*π*) using DnaSP [50]. Additionally, we calculated the average number of MHC alleles per site (*An*) across hamsters.

To investigate the phylogenetic relationships among these alleles, we compiled 20 MHC-DRB II alleles of striped hamsters from the Shandong Provinces (Genbank: HM102423~HM102442). We constructed a neighbor-net network using SplitsTree [51] to visualize potential reticular relationships. We selected the JC+G model of nucleotide substitution based on the AIC values obtained from JmodelTest. The neighbor-net network was constructed with edge weights using ordinary least squares variance and a threshold of 10.

We conducted Bayesian Clustering analysis in STRUCTURE using correlated allele frequencies, without incorporating prior location information. For each K value, we performed ten independent Markov Chain Monte Carlo (MCMC) runs, each consisting of 1,000,000 generations with a burn-in of 500,000. The input file for structure analysis assigned a value of 1 if the corresponding alleles were present for each individual, 0 if they were not present, and −9 if the data was missing. To further examine the population structure exhibited at the MHC DRB locus, we performed Principal Component Analysis (PCA) to maximize the differences between populations and minimize the variations within populations. The inter-population differentiation at the MHC locus was quantified using pairwise Rho values. To consider nonlinear population distribution and migration patterns, the pairwise Rho values were transformed to Rho/(1-Rho). The analyses were conducted using SPAGeDi [52]. 

### 2.8. Comparisons between Neutral and Adaptive Markers

To compare the substructuring levels exhibited by two different marker types, we conducted Mantel tests and partial Mantel tests using the Vegan package in R [53]. To examine patterns of isolation by distance, we conducted separate tests to determine the relationship between microsatellite loci and geographic distance, and between the MHC-DRB locus and geographic distance. Geographical distances were calculated as Euclidean distances using the pointDistance function in the RASTER R package [54,55]. Additionally, we conducted further tests for isolation by distance at the MHC-DRB locus by comparing Rho/(1-Rho) values for MHC with geographic distance, while accounting for neutral pairwise differentiation. Finally, we assessed the correlation between pairwise estimates of Rho/(1-Rho) for microsatellites and MHC-DRB, while considering geographic distance. The significance of all correlations was evaluated using 999 permutations. 

Population genetic differentiation at MHC loci is expected to be more pronounced than that resulting from neutral genetic variation due to selective pressure. We chose to perform a co-inertia analysis (CoA) [56] to assess the relationship between the MHC gene and microsatellites. Due to its advantages over traditional genetic differentiation methods and its avoidance of equilibrium assumptions, CoA was deemed beneficial for assessing the genetic co-structure between MHC and microsatellites, and for inferring patterns of local adaptation. The PCA results of MHC and microsatellite data were used as input files to construct the CoA. In a CoA plot, the length of the vector connecting the arrow and the dot represents the divergence between the marker types. If both genetic markers exhibit strong shared trends, the arrow will be short.

## 3. Results

### 3.1. Parasite Communities

The ectoparasite burden of these four study populations was shown in Table 1. The flea load ranged from 0 to 17, while the gamasid mite load ranged from 0 to 38. We included a total of 11 flea species and 19 gamasid mite species (Appendix A). In the W4 population, we observed one individual carrying a maximum of five flea species, and the hamster infected with up to five gamasid mite species was also from W4. Among the four analyzed populations, W4 exhibited the highest species richness for both fleas (0.49) and gamasid mites (1.09). W2 had the highest flea load (1.19), while W4 had the highest average number of gamasid mites (2.89). Moreover, the highest infection prevalence of gamasid mite was 55% in W4, and the highest flea infection rate was 47% in N2. We observed that the load and abundance of gamasid mites in the N2 population were significantly lower than those in the other three populations. We found no significant difference in flea burden among these populations (Figure 2).

Temperature increase was positively correlated with gamasid mite load (0.175 ± 0.038, *p* < 0.05), gamasid mite richness (0.131 ± 0.060, *p* < 0.05), and flea load (0.241 ± 0.067, *p* < 0.05). Precipitation was positively correlated with gamasid mite burden (load: −0.004 ± 0.001, *p* < 0.05; richness: −0.004 ± 0.001, *p* < 0.05). However, we did not observe any significant effect of precipitation on flea burden (load: −0.001 ± 0.001, *p* = 0.19; richness: −0.002 ± 0.001, *p* = 0.114), as shown in Figure 3.

### 3.2. Microsatellite Genetic Diversity and Population Structure

This study utilized 7 pairs of fluorescent primers to amplify 181 samples from 4 populations, resulting in polymorphic loci for all markers. The overall polymorphic characteristics of microsatellite markers were summarized in Table 2. The *Na* varied among populations, ranging from 6.094 (N2) to 21.429 (W4). The *Ne* in all populations was lower than the *Na*, with an average value of 8.781. The lowest *Ho* was 0.477 at N3, while the highest was 0.600 at W4. Observed heterozygosity was higher than expected heterozygosity at all sites. The HWE test indicated that all populations were in equilibrium, except for W4. No significant departures from LD were observed within global regions after Bonferroni correction. 

Although the Bayesian clustering analysis with STRUCTURE identified the most likely number of clusters as 2, there was no significant population genetic structure observed among the populations (Figure 4A). In the PCA plot (Figure 5A), the first axis (65.63%) clearly separated N2 from the other populations. Pairwise Rho/(1-Rho) values are presented in Table 3. The highest pairwise Rho/(1-Rho) value among the four populations was observed between N2 and W4 (0.029). 

### 3.3. MHC Genotyping

The average depth of individual sequencing coverage for MHC DRB exon 2 was 4381 reads (ranging from 3331 to 4826). There was no significant correlation between the number of alleles and sequencing depth (*R*^2^ = 0.001, *p* = 0.733), indicating that our sequencing was adequate for reliable genotyping. Our validation process identified a total of 89 unique sequences. The MHC01 allele was the most prevalent among nearly all individuals. 

### 3.4. MHC Selection

The Z test revealed a significant excess of non-synonymous substitutions, indicating a high overall ω (*dN*/*dS*) ratio for residues (Z = 2.67, *p* < 0.05). Positive selection was detected in all four nested models based on likelihood ratio test (LRT) *p* values (Appendix A). The M8 model identified three codons (site 5, 16, and 36) under positive selection. Using the FEL, MEME, FUBAR, and SLAC site mutation models in DataMonkey, we identified 11 out of 57 sites that were considered positively selected (Table 4). Among these, two sites were found to be under positive selection across all four models, while the remaining sites showed positive selection in less than three tests. Additionally, no recombination events were detected as there were no significant potential breakpoints identified by the GARD recombination algorithm.

### 3.5. MHC Population Diversity and Structure

Multiple MHC loci in the striped hamsters contain the sequence, as evidenced by the detection of up to 25 alleles within an individual. The majority of individuals carried 7–15 alleles, with only two individuals having fewer than 3 alleles. Among the sampled locations, W4 exhibited the highest level of diversity with a total of 70 alleles, whereas N2 had the lowest diversity. DRB exon 2 displayed a substantial average nucleotide diversity (*π* = 0.105). The number of segregating sites (*S*) varied from 66 to 77 across populations, and the haplotype diversity (*Hd*) exceeded 0.97 for all populations (Table 5). 

The split network analysis of 119 alleles did not show significant clustering patterns indicative of the trans-population evolution of MHC-DRB loci, as depicted in Figure 6. The alleles from different sites exhibited a high degree of similarity in the striped hamster. Notably, only one sequence (MHC10) was found in both Inner Mongolia and Shandong (HM102432), and the alleles from these two regions were dispersed across the network with low bootstrap values.

In the STRUCTURE analysis of the MHC binary-encoded data, the highest likelihoods were obtained when the samples were clustered into 2 groups. Similar to the microsatellite analysis, the MHC alleles did not exhibit clear subdivision between different populations (Figure 4B). The between-class PCA analysis of the MHC revealed that the first axis, accounting for 51.91% of the variation, distinguished N2 from the other populations. Furthermore, pairwise Rho values based on the MHC data supported the absence of genetic differentiation between the locations (Figure 5B). 

### 3.6. Comparing MHC and Neutral Diversity

There was no significant pattern of isolation by distance observed for microsatellites (*r* = 0.00, *p* = 0.667) or MHC-DRB (*r* = −1.00, *p* = 1.00). Controlling for neutral diversity, the correlation between the MHC-DRB locus and geographic pairwise distances remained non-significant (*r* = −1.00, *p* = 1.00). Additionally, no association was found between MHC-DRB and microsatellite inter-population diversity when controlling for geographic distance. Despite the dissimilarity in cluster results between PCA-based microsatellite and MHC, the CoA plot (Figure 5C) revealed overlapping co-structure between the MHC and microsatellite, indicating a strong correlation between them.

## 4. Discussion

In this study, we investigated the spatial genetic variation of adaptive major histocompatibility complex (MHC) genes in relation to neutral microsatellite loci across four populations of striped hamsters in Inner Mongolia. We observed significant variation in parasite pressure among sites, with parasite burden showing correlation with temperature and precipitation. Molecular analysis revealed a similar co-structure between MHC and microsatellite loci. We observed lower genetic differentiation at MHC loci compared to microsatellite loci, and no correlation between the two. Our findings suggest that MHC genetic variation is primarily influenced by neutral processes, with a lesser contribution from balancing selection, and no evidence of local adaptation.

### 4.1. Parasite Diversity at Different Sites

Despite extensive research devoted to studying parasite burden, our understanding of parasite distribution remains limited. In this study, we observed significant variation in gamasid mite burden among different sites, indicating that environmental heterogeneity may influence gamasid mite distribution. However, there was no significant difference in flea burden, possibly due to a dilution effect caused by a high number of uninfected host individuals in our study. It should be noted that, since we used clips to catch striped hamsters, some fleas migrated to find new hosts after the hamsters died, despite our efforts to minimize sampling time. Nevertheless, this migration had a consistent impact on all individuals and did not significantly affect our results. 

We observed a significant correlation between the annual temperature levels at each sampling site and the parasite burdens found in striped hamsters. Specifically, we noted that flea load, gamasid mite load, and gamasid mite richness increased with higher temperatures. Climate factors can influence parasite development and transmission [57], abundance [58], and richness [59]. In general, warmer temperatures can enhance parasite growth and reproduction rates, leading to higher parasite burdens in hosts. Several studies have explored the relationship between temperature and parasite burden [60,61]. Samuel et al. reported that temperature increases were associated with an increase in off-host flea abundance, and they predicted that burrow temperatures and flea development rates would rise, potentially resulting in higher flea abundance [62]. Hammond et al. also demonstrated a notable connection between ambient temperature variation and the abundance of two flea species [63]. 

Precipitation can have significant effects on parasite pressure. This is because the free-living stages of parasites are directly exposed to the environmental conditions in their respective microhabitats [64,65]. Our findings suggest that precipitation may decrease gamasid mite burden. Heavy precipitation could inhibit the population growth of gamasid mites in this region. A study conducted in Yunnan, China, also demonstrated different effects of precipitation on various mite species [66]. In contrast, conflicting results have been reported regarding the influence of precipitation on parasitic load. For instance, in the spruce forests of Eastern Europe, regions with high rainfall volume and well-developed moss sod exhibit the highest populations of gamasid mites [67]. However, we did not observe any significant effect of precipitation on flea burden. A study from Vietnam, adjacent to southern China, showed a decreasing exponential trend in the flea index with increasing monthly average rainfall [68]. Experimental work has also shown that relative humidity significantly affects flea survival [69]. The effects of precipitation on fleas and gamasid mites differ depending on their ecological habitats. For example, compared to the more climatically variable habitats of the fynbos and the highland forests in Africa, greater parasite diversity was found in lowland rainforests [70]. A study investigating infestation levels of ectoparasites on rodents found higher ectoparasite indices in scrub habitats than in forest and farmland habitats [71]. When the habitat is homogeneous, the host composition is also homogeneous, resulting in lower parasite richness. Habitat type affects parasite diversity mainly by altering the distribution and composition of the hosts. When the habitat is homogeneous, the host composition is also homogeneous and thus, the parasite richness is low. Therefore, the effect of precipitation on parasites can be complex and depends on various factors. On the one hand, rainfall can increase the humidity in the environment, creating ideal conditions for the survival and reproduction of some parasites. On the other hand, heavy rainfall can have a negative effect as flooding can wash away parasite habitats and disrupt their life cycles.

### 4.2. Characterization of MHC Polymorphism, Historical Selection, and Parasite Resistance

The MHC represents one of the most genetically variable regions in vertebrate genomes. Most species that have been investigated exhibit high levels of MHC diversity [72,73], which is crucial for individual fitness and the adaptive potential of population responses to environmental pressures [74]. In this study, we observed a high allelic richness among striped hamster populations. With a total of 89 alleles, MHC DRB exon2 exhibited high diversity. This level of MHC diversity was relatively higher compared to a previous study conducted in Shandong Province, where only three or four alleles were obtained from five clones for each individual [75]. To the best of our knowledge, this is the first study to report MHC variation in striped hamsters using a high-throughput sequencing method. Compared to traditional methods, this approach significantly improves sequencing accuracy, enabling the detection of rare and poorly amplified MHC variants [73,76]. Studies conducted on other rodent species (15 to 21) have reported similar numbers of MHC loci compared to those identified in our study [77]. However, fewer gene copies have been detected in certain mouse species [18,78,79]. The mechanisms underlying the observed high levels of polymorphism in MHC genes remain unclear. It is believed that MHC maintains high genetic diversity through pathogen-mediated selection [80,81]. Theoretically, heterogeneous individuals are favored under balancing selection, which leads to enhanced immune responses against pathogens. A study involving six experimental populations exposed to different parasites demonstrated an increase in genes providing resistance to specific parasites in the subsequent generation, indicating that rapid adaptive evolutionary changes occur due to varying parasite selection, thereby facilitating the maintenance of MHC polymorphism [82]. We discovered a prominent indication of historical positive selection, with ω > 1, in the peptide binding regions of MHC loci, suggesting a higher rate of non-synonymous substitutions than expected under neutral selection. Additionally, by employing a series of codon models, we identified several positive selection sites within the MHC molecules. It is plausible that micro-recombination, to some extent, contributed to the diversity of MHC-DRB sequences, as evidenced by the presence of multiple circular patterns connecting nodes on the splits tree, despite not detecting recombination in our sequences [83]. 

### 4.3. Neutral Processes Mostly Shaped MHC Population Variation

The ability of populations to adapt across spatially heterogeneous and temporally variable landscapes is influenced by the competing forces of natural selection, gene flow, and genetic drift [2]. Although demonstrating patterns of local adaptation proved challenging, it is possible to distinguish the effects of selection from those of gene flow and genetic drift by comparing variations at adaptive and neutral loci [84,85,86]. In this study, we have found that the pattern of MHC-DRB variability to align with neutral forces: MHC-DRB exhibited a similar population structure to that of microsatellite loci based on CoA analysis. This indicated that, at the population level, neutral processes played a more significant role than balancing selection in driving the spatial variation of MHC [17,87]. Among non-selective forces, genetic flow has played a significant role in determining the observed differentiation and diversity at the MHC. This is supported by several lines of evidence: (i) the four striped hamster populations exhibit panmixia with no genetic structure detected at the MHC, suggesting substantial gene flow between populations that mitigates the effects of genetic drift. Furthermore, (ii) no apparent isolation-by-distance was observed for both MHC and microsatellite divergence.

In contrast to the genetic structure observed in microsatellite loci, MHC-DRB exhibited lower genetic differentiation among the four populations, indicating weaker genetic structure. Pairwise Rho/(1-Rho) values for microsatellites displayed greater magnitude than those for MHC, and no correlation was found between them. This disparity may stem from limitations in uncovering patterns of genetic structure when transforming data into binary-encoded format. Nevertheless, this approach does offer a means to compare multi-locus MHC and microsatellite data [88]. Indeed, reduced genetic structure relative to neutral loci has been documented in other animal systems, including wolverine [89], island foxes [90], and zebras [90]. It has been proposed that the genetic similarity across populations is attributed to homogenizing directional selection driven by a common parasite affecting all populations. Based on these findings, we suggest that, to some extent, balancing selection may influence the genetic variation of striped hamsters, despite the predominance of neutral processes.

### 4.4. The Spatial Scale of Local Adaptation

Gene flow among the four populations is expected due to the relatively small geographical scale of this study, in contrast to the extensive range of the striped hamster throughout northern Asia. It is noteworthy that, although we did not observe any local adaptation of MHC in Inner Mongolia, the MHC allele frequencies of striped hamsters in this region differ from those in Shandong Province. This suggests that patterns of local adaptation may vary depending on the spatial scale. While local adaptation has been frequently demonstrated, our understanding of its spatial scale remains limited. Generally, the prevalence of local adaptation tends to increase with geographical distance [91]. As the geographical distance increases, greater genetic isolation and environmental differentiation are expected. Several studies have shown that local adaptation occurs primarily at larger spatial scales (between regions), rather than among populations within regions [92,93]. When the homogenizing effect of gene flow is limited and divergent selection consistently operates, local adaptation can occur on micro-geographical scales, particularly in small, isolated populations [94]. The balance between gene flow and the strength of natural selection is crucial for local adaptation. The solitary lifestyle of the striped hamster may also influence local adaptation. Studies have found that the African striped mouse (*Rhabdomys pumilio*) can switch from group to solitary living with the onset of the breeding season to avoid competition for reproduction within groups [95]. Additionally, males are the dispersing sex compared to females, and less competitive males tend to migrate longer distances in search of reproductive success. The sex ratio of the striped hamster is close to 2:1, indicating that males face greater pressure to reproduce successfully. Although no studies have confirmed migration in striped hamsters, we hypothesize that this trait facilitates gene flow between populations and partly contributes to the limited occurrence of local adaptation in striped hamsters under heterogeneous pressures at broader spatial scales.

### 4.5. Implications for Plague Control

Four plague foci are located in Inner Mongolia: the *Meriones unguiculatus* plague focus in the Inner Mongolian Plateau, the *Microtus brandti* plague focus in the Xilin Gol Grassland, the *Spermophilus dauricus* plague focus in the Song-Liao Plain, and the *Marmota sibirica* plague focus in the Hulun Buir Plateau. Recent years have seen reported cases of plague in this region [95]. Previous studies have identified the striped hamster as a host of plague bacteria in all of the aforementioned areas. The striped hamster can be infected with various fleas, some of which are important vectors of plague, such as *Citellophilus tesquorum mongolicus*, *Amphipsylla primaris mitis*, *Frontopsylla luculenta*, and others [96]. Based on the findings of this study, frequent gene exchange and consistent adaptive traits were observed among different populations of striped hamsters. Therefore, the most crucial step that public health authorities in Inner Mongolia can take is to monitor inter-animal plague transmission and control the density of rodent populations, particularly in those plague foci.

## 5. Conclusions

Our findings indicate that the genetic variation at MHC DRB exon 2 in striped hamsters is influenced by the interplay of neutral processes (gene flow) and, to a lesser extent, balancing selection. Additionally, we emphasize the importance of considering spatial scale in investigations of local adaptation. Collectively, this study contributes to our understanding of MHC gene evolution in wild populations. Furthermore, given the significant gene flow observed, efforts aimed at preventing the spread of plague should prioritize monitoring inter-animal plague transmission and controlling rodent density, particularly in areas affected by plague outbreaks.

## Figures and Tables

**Figure 1 genes-14-01500-f001:**
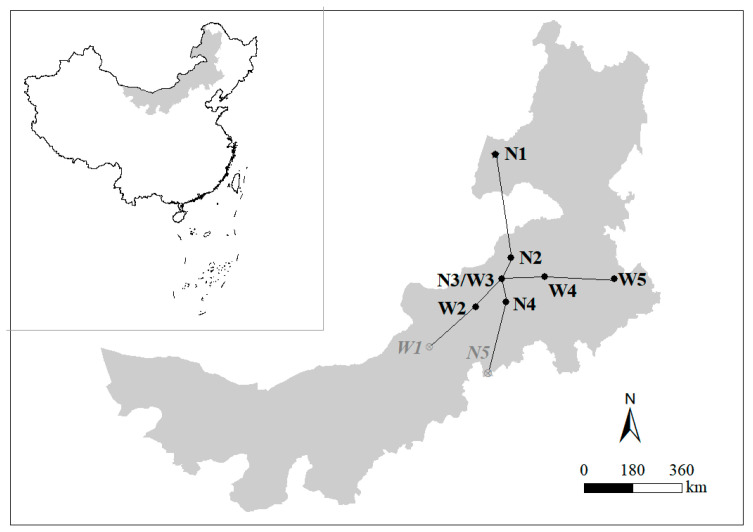
Sample locations in Inner Mongolia. The sample sites are located in New Barag Right Banner (N1), East Ujimqin Banner (N2), Xilin Hot (N3/W3), Baiyin Siler (N4), Taipusi Banner (N5), Taibus Banner (W1), Sonid Right Banner (W1), Abag Banner (W2), West Ujimqin Banner (W4), and Tongliao (W5). The black dots represent locations where striped hamsters were sampled in this study.

**Figure 2 genes-14-01500-f002:**
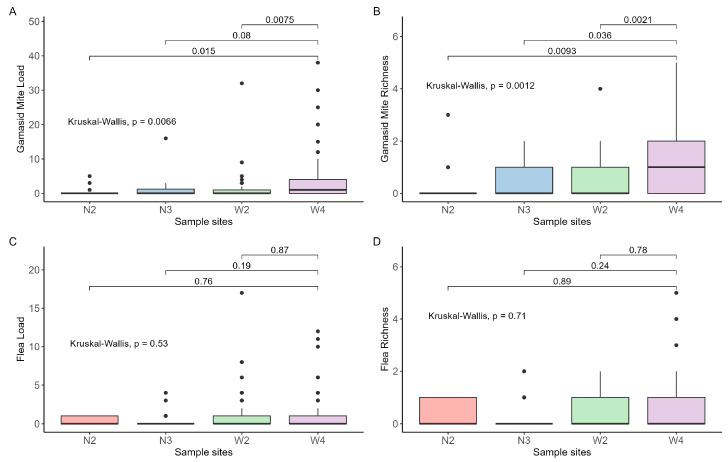
Parasite burden among populations: (**A**) gamasid gite load, (**B**) gamasid mite richness, (**C**) flea load, and (**D**) flea richness.

**Figure 3 genes-14-01500-f003:**
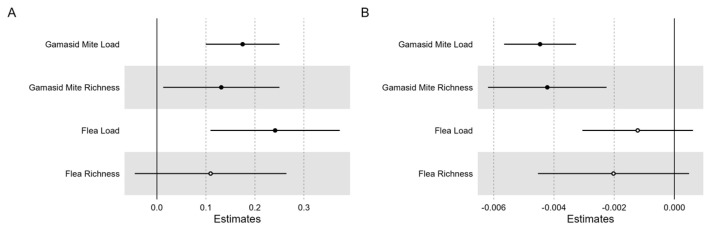
Model−averaged parameter estimates and their 95% confidence intervals for (**A**) temperature and (**B**) precipitation associated with four parasite indices. ● a parameter with a significant effect.

**Figure 4 genes-14-01500-f004:**
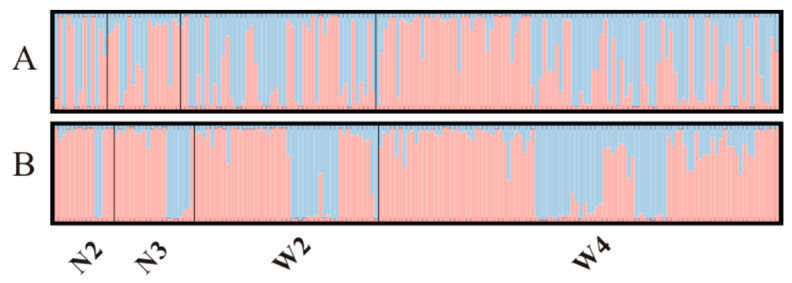
Population genetic structure of striped hamsters estimated from microsatellite (**A**) and MHC allelic data (**B**). The population structure estimated in structure using the most reliable number of clusters (K = 2). Each vertical bar represents an individual. The height of each bar indicates the probability of assignment to each of K optimal clusters (different colors).

**Figure 5 genes-14-01500-f005:**
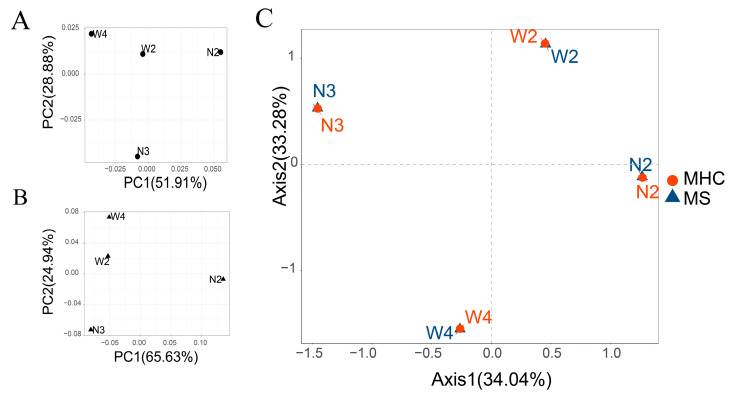
Co−inertia analysis (CoA) between microsatellite and MHC binary−encoded data for seven regions. Ordination of the first two between-class axes for (**A**) microsatellite and (**B**) MHC loci, where dots represent populations constrained by sampling locations distinguished in different colors. (**C**) CoA plot, showing the relative position of each population on the factorial plane for the first two CoA eigenvalues and given by the co-variation between MHC and microsatellite data sets. The red dots represent the variation observed at MHC, while the blue triangles represent the variation at microsatellite (MS).

**Figure 6 genes-14-01500-f006:**
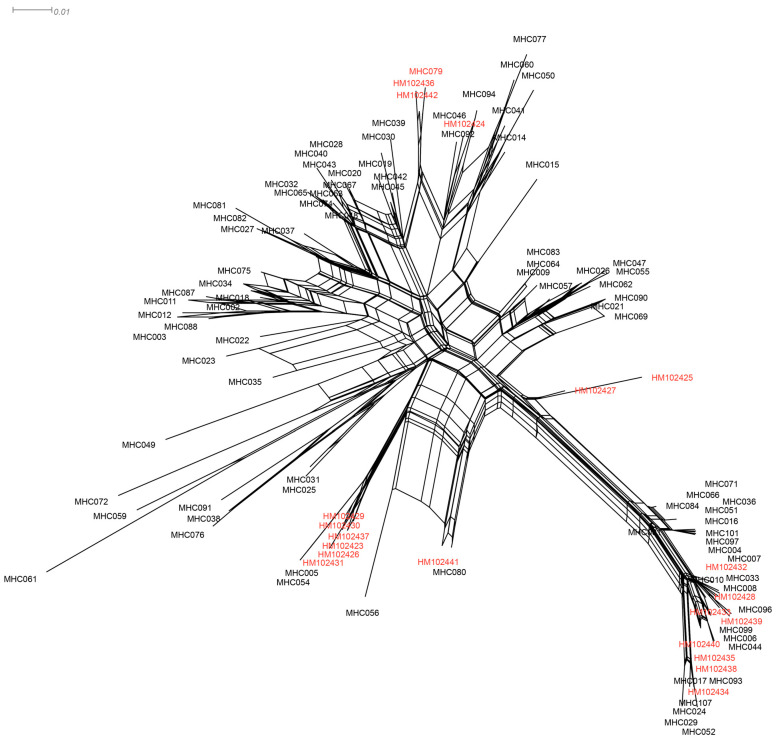
Neighbor-net network of striped hamster MHC-DRB alleles constructed from alleles detected in this study and those previously reported in the literature. The loops imply areas of phylogenetic uncertainty or reticulations. The alleles marked with red were detected in Shandong.

**Table 1 genes-14-01500-t001:** Ectoparasite burden (load, richness, and prevalence) and climatic variables of these four study populations.

	N2	N3	W2	W4
Gamasid Mite Load	0.59	1.29	1.32	2.89
Flea Load	0.47	0.42	1.19	0.92
Gamasid Mite Richness	0.31	0.45	0.47	1.09
Flea Richness	0.35	0.29	0.40	0.50
Gamasid Mite Prevalence	24%	38%	37%	55%
Flea Prevalence	47%	21%	34%	35%
Annual Mean Temperature	1.78	2.14	7.18	4.98
Annual Precipitation	242	249	440	201

**Table 2 genes-14-01500-t002:** Genetic diversity in seven populations of striped hamster from Inner Mongolia, China.

Pop	*N*	*Na*	*Ne*	*I*	*Ho*	*He*	*uHe*	*F*	*Ar*	HWE
N2	17	9.286	6.094	1.975	0.496	0.828	0.862	0.390	4.84	0.167 ^ns^
N3	24	10.571	7.708	2.116	0.477	0.850	0.874	0.428	5.02	0.092 ^ns^
W2	62	15.143	9.176	2.362	0.552	0.873	0.883	0.358	5.15	0.104 ^ns^
W4	111	21.429	12.146	2.594	0.600	0.890	0.894	0.317	5.34	0.000 *
Total	214	14.107	8.781	2.262	0.531	0.860	0.878	0.373	5.08	/

Abbreviations: *N*, number of samples; *Na*, number of alleles per population; Ne, effective allele; *I*, Shannon’s information index; *Ho*, observed heterozygosity; *He*, expected heterozygosity; *uHe*, unbiased expected heterozygosity; *F*, fixation index; *Ar*: rare allelic richness; HWE: Hardy–Weinberg equilibrium test, significant (*p* < 0.05) deviations from Hardy–Weinberg expectations are indicated with an asterisk. Key: ^ns^ = not significant, * *p* < 0.05.

**Table 3 genes-14-01500-t003:** Population pairwise Rho/(1-Rho) values for seven microsatellite loci (lower left) and for MHC-DRB (top right) in four striped hamster populations.

N2	N3	W2	W4	
	0.006	0.005	0.043	N2
0.014		0.004	0.020	N3
0.020	0.003		0.022	W2
0.029	0.018	0.016		W4

**Table 4 genes-14-01500-t004:** Identification of the codons in MHC-DRB.

Models	5	9	16	17	23	30	36	40	46	49	53	56	57
FEL									√				√
MEME					√		√	√	√	√	√	√	√
FUBAR		√		√		√			√	√			√
SLAC		√		√					√				√
M8 vs. M7	√		√				√						

**Table 5 genes-14-01500-t005:** Genetic variation at MHC-DRB for four striped hamster populations from Inner Mongolia.

Pop	*h*	*S*	*Hd*	*K*	*π*	*An*
N2	47	66	0.972	18.49	0.108	10.80
N3	51	67	0.971	18.12	0.106	10.45
W2	66	76	0.971	18.01	0.105	11.91
W4	70	77	0.970	17.86	0.104	12.52

Abbreviations: *h*, number of haplotype per population; *S*, number of segregating sites; *Hd*, haplotype diversity; π, mean nucleotide diversity; *An*, the average number of MHC variants (per hamster) in each site.

## Data Availability

Data supporting the conclusions of this article are included within the article and its Appendix A.

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
