# Peer review of "Neutral Forces and Balancing Selection Interplay to Shape the Major Histocompatibility Complex Spatial Patterns in the Striped Hamster in Inner Mongolia: Suggestive of Broad-Scale Local Adaptation"

_genes, 2023, doi:10.3390/genes14071500_

Round 1
Reviewer 1 Report
The study titled “Neutral forces, and balancing selection interplay to shape MHC 2 spatial-patterns in striped hamster in Inner Mongolia: suggestive of broad scale local adaptation” looks at the relative strength of neutral forces vs selection acting on MHC variation in this rodent species, working with four wild populations which differ in levels of average ectoparasite burden. The work is well written and sound but the following comments must be addressed:
- Introduction: Overall, this section is clear and well-written, but predictions should be formulated at the end of the introduction in order to guide the reader across the rest of the manuscript. Further, in the discussion, authors mention that they “expected the pattern of MHC variability to align with neutral forces”, so this prediction should be made explicit, as well as others. See comment below.
- Methods: Line 185. Please provide data on how frequent were these non-functional alleles in the sample.
- Results: Line 270-272 and throughout the text, the terms “burden”, “load” or even “parasite stress” (abstract line 17) are used as synonyms. Please, unify the use as it is confusing.
- Line 290: Please include sample size per population in Table 1
- Also, please summarize in a table the data on ectoparasite load, richness (name the different ectoparasite species in the supplementary material) and climatic variables of the four study populations. And add the calculations of infection prevalence (proportion of hosts infected), which is also a key variable to understand levels of parasite exposure across populations.
- Line 326: Were these sequences deposited in Genbank? If not, how are these sequences going to be made available after publication?
- Line 472: why did authors expect the pattern of MHC variability to align with neutral forces? In the introduction, it is said that “the striped hamster serves as an ideal model for investigating local adaptation mediated by parasites” (line 81). Also, there is evidence that suggests that local adaptation may be plausible in this species (Line 79)
-
- Aside from several studies cited on MHC variation in fish, the studies on rodents cited date from 2014 and back. Please, include the following more recent references of studies on mammals with very similar questions and methods to the present work:
Del Real-Monroy and Ortega. 2017. Mammalian Biology (Jamaican fruit bat)
Cutrera and Mora. 2017. Journal of Heredity (subterranean rodent)
Buzan et al. 2022. Animals (roe deer)
Kong Lam et al. 2023. Evolution (European badger)
Minor comments:
- Abstract, line15: Please check sentence writing: “… on MHC population and diversity…”
- Legend of figure 5: Plots A and B do not show different colors to distinguish sampling locations
- Please improve quality of figure 6, it is very hard to read.
Only minor editing is needed (as suggested in comments to authors)
Reviewer 2 Report
I support the publication of this manuscript as it ism but I have minor comment which should be corrected before publication. The reference under the N27, line 633, page 15 erroneously starts with authors first names instead of last names. Natalia, P. should be changed to Poplavskaya, N. and all other authors in the same manner respectively.
Reviewer 3 Report
The manuscript deals with to shape MHC spatial-patterns in striped hamster in Inner Mongolia. The aim and content of manuscript are interesting. Introduction is well written. The results of the research obtained have been properly described and discussed and provide some new findings.
Comments:
In Material and methods
- it is not mentioned how the libraries were quantified (Quibit, qPCR)
In Discusion
- in lines 428-429, 431-432 there are sentences with the same meaning
